# ALIGNING LARGE LANGUAGE MODELS WITH PREFERENCE PRIVACY

## ABSTRACT

Alignment is a crucial part in the implementation pipeline of Large Language Models (LLMs) that utilizes human feedback to ensure that LLMs adhere to human values and societal norms. This introduces privacy threats associated with the identity and preferences of the labelers responsible for creating the human feedback data. Several recent works have explored using differential privacy (DP) as a notion to protect the privacy of human labeled data; primarily relying on DP-SGD based solutions, which privatize the gradients during fine-tuning and alignment. Human preferences, however are only associated with the labels of the (prompt, response) tuples; therefore DP-SGD based approaches can be superfluous, providing more privacy than necessary and can degrade model utility. In this work, we focus on the problem of aligning LLMs with preference level privacy, which only preserve the privacy of preferences provided by humans. We build and expand upon the concept of label DP for this problem, and present a series of increasingly sophisticated, yet practical privacy preserving mechanisms for alignment. Specifically, starting from a standard randomized response (RR) mechanism which randomly flips human preferences, and its corresponding *unbiased* RR mechanism (which ensures an unbiased loss during alignment), we propose a new mechanism, PROPS (PROgressively Private Self-alignment). PROPS works in multiple stages as follows: in each stage, the privately trained and partially aligned model from the previous stage to act as a labeler for the training data for the next stage and combine it with RR which is repeated across multiple stages. Motivation for PROPS comes from the following critical observations: a) learning to label correct preferences might be an easier problem than generating responsible content; b) progressively combining RR with partially aligned models for labeling preferences significantly reduces the amount of necessary perturbation needed for privacy and also shows the potential of possibly reducing the number of human labeled preference samples. We present comprehensive experiments using multiple models (including Pythia and GPT models), and two datasets (`truthy-dpo-v0.1` and Anthropic HH-RLHF) to showcase the utility gains of PROPS over existing privacy preserving methods (including DP-SGD and RR). Our results demonstrate that PROPS is especially effective in high-privacy regimes compared to conventional DP-SGD. For example, with $\epsilon = 0.1$ the Win-Tie-Lose rates of PROPS against DP-SGD for GPT2-Large and GPT2-Medium are respectively $66 : 2 : 32$ and $76 : 1 : 23$, which demonstrate a clear advantage of using PROPS.

## 1 INTRODUCTION

In recent years, Large Language Models (LLMs) have gained significant attention across various disciplines due to their ability to generate responses based on open-ended human instructions. Training LLMs typically involves three key stages: pre-training, supervised fine-tuning (SFT), and alignment. Among these, alignment is particularly important as it guides LLMs to produce responses that align with societal norms and human preferences. The alignment process typically relies on a dataset $\mathcal{D}$ consisting of $n$ samples, each containing a prompt $x$, LLM-generated responses $(y_1, y_2)$, and a human-preferred label $\ell^*$, collectively referred to as preference data (which one of $y_1$ or $y_2$ is the prefered response). Over the past few years, two main alignment approaches have emerged, namely, Reinforcement Learning with Human Feedback (RLHF) (Stiennon et al., 2020) and Direct Preference Optimization (DPO) (Rafailov et al., 2024), both of which utilize preference datasets

with ranked labels provided by human annotators. While alignment improves the quality of generated responses, the use of human-labeled samples introduces privacy risks, particularly concerning the identity of the annotators and their associated preferences. Intuitively, LLM generated (prompt, response) pairs in datasets may not require strict privacy protections, but human preference labels require privacy, as they often reveal sensitive insights. For instance, in medical applications, publicly available case reports or anonymized symptoms can prompt LLMs to generate diagnoses, with human feedback providing expert alignment with best practices, necessitating privacy to safeguard clinical judgments. Similarly, in policy analysis, publicly available survey questions or proposals can elicit LLM-generated analyses, where policymakers' feedback reveals sensitive interpretative insights that may require protection.

***Related works & motivation:*** To mitigate the privacy risks associated with human-annotated preference data, the notion of Differential Privacy (DP) (Dwork et al., 2014) has recently been explored for fine-tuning and alignment of LLMs. For example, Yu et al. (2021) applied DP to fine-tuning by introducing privacy guarantees for smaller, appended parameters such as LoRA and adapters. Behnia et al. (2022) proposed a DP fine-tuning framework for LLMs using the Edgesworth accountant, while Zheng et al. (2024) provided DP guarantees for in-context learning. Singh et al. (2024) introduced a two-stage fine-tuning process, and Yu et al. (2024) addressed the privacy-preserving alignment challenge by ensuring

| Privacy | PROPS vs DP-SGD (Win-Tie-Lose) | |
| Budget | GPT2-Large | GPT2-Medium |
| --- | --- | --- |
| $\epsilon = 0.1$ | **66** : 2 : 32 | **76** : 1 : 23 |
| $\epsilon = 0.5$ | **60** : 2 : 38 | 50 : 11 : **39** |
| $\epsilon = 1$ | **60** : 3 : 37 | 49 : 7 : **44** |
| $\epsilon = 2$ | 44 : 6 : **50** | **54** : 4 : 42 |

Figure 1: Comparison of Win-Tie-Lose rate for PROPS vs DP-SGD for high privacy regimes with GPT2-Large, GPT2-Medium models (for more results see Section 4).

DP protection for users' prompts against labelers during the generation of preference datasets for alignment. DP for LLM inference privacy has also been studied by Mai et al. (2024) and Flemings et al. (2024). Additionally, Wu et al. (2023) proposed applying DP to RLHF by splitting the dataset into three disjoint sets to ensure DP at each stage of RLHF. We now make the following critical observations that motivate this paper:

1. The aforementioned works view the entire tuple of $\{\text{prompt} = (x)\text{ , responses} = (y_1, y_2),$ human-annotated labels $= (\ell^*)\}$ as a private entry in the training dataset and the associated DP notion is invoked to protect this entire tuple. We note that in typical alignment scenarios, the prompts and responses are not generated by humans (the two responses $(y_1, y_2)$ are usually obtained from the fine-tuned LLM). Human input is only used to label/rank the responses. Therefore, providing privacy of the entire tuple can be superfluous and potentially hurt the *utility* of the privately aligned model. This observation motivates the problem of this paper: alignment with preference privacy, where only the preferences of the labelers need to be protected, rather than prompt and response pairs.

2. Most of the existing works listed above achieve DP by using DP-Stochastic Gradient Descent (DP-SGD) and their variants, such as Differentially Private Proximal Policy Optimization (DP-PPO). These approaches modify the training procedure by privatizing the gradients based on the leakage budgets. While these methods are adequate in terms of privacy, they provide the privacy for the entire tuple (i.e., lack the specificity of only keeping human preferences private) and often degrade performance in high-privacy regimes (as shown in Figure 1).

***Overview and Main Contribution:*** Motivated by the above observations, we study the problem of aligning LLMs with preference privacy. Specifically, we investigate two notions of privacy: a) preference-level privacy and b) a stronger notion of labeler-level privacy. Preference level privacy ensures that the individual human-preference $\ell^*$ for any tuple $(x, y_1, y_2)$ does not significantly impact the aligned model. Formally, we leverage the existing notion of Label-DP, and use it to formalize the notion of $(\epsilon, \delta)$-preference-level DP, where $(\epsilon, \delta)$ represent the privacy budgets. The stronger notion of labeler-level privacy (also commonly referred to as "user" level privacy in the DP literature) hides the presence/absence of any individual human labeler and protects all the labels annotated by the labeler. For the scope of this paper, we focus predominantly on DPO Rafailov et al. (2024) as the alignment algorithm for two reasons: a) there is no existing work on DPO with differential privacy; and b) DPO is less computationally expensive than RLHF, since DPO learns a

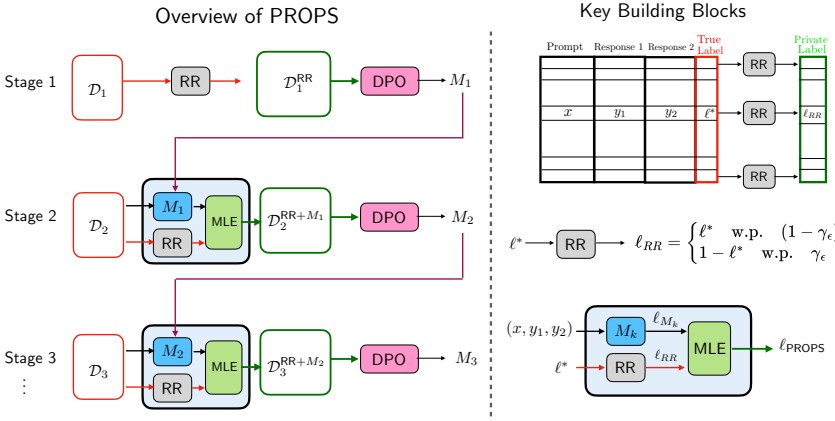

Figure 2: PROPS: *Progressively Private Self-Alignment* algorithm: PROPS aligns in multiple stages, where in each stage, the model learned from the previous stage is used to generate labels/preferences for the next batch; these preferences together with the noisy versions (perturbed via randomized response (RR)) of ground truth preferences are combined via a maximum-likelihood estimator (MLE) which are used for alignment.

aligned model directly, whereas RLHF involves training a reward model, followed by reinforcement learning stage. We summarize and highlight the main contributions and novel aspects of this work:

- *Randomized Response base methods*: We start by studying the classical mechanism of Randomized response (RR) which directly perturbs the human-labels, which can then be used for model alignment via conventional DPO training. This method however, ends up introducing a bias in the loss.

- *PROPS Algorithm*: we present *Progressively Private Self Alignment (PROPS)*, a novel algorithm which takes a more nuanced approach to alignment through a multi-staged process. Rather than using the entire perturbed dataset in one go, PROPS divides the alignment task into multiple stages, each involving a fraction of the dataset (see Figure 2). Let us focus on the $k$th stage for $k > 1$: at this stage we have a model obtained from the previous stage, denoted by $M_{k-1}$. We now make two observations: in the $k$th stage in order to use fresh batch of samples, we still need to perturb the true private labels (say using RR). However, we also have access to the model $M_{k-1}$ (which can be used without additional leakage due to post-processing of DP), as well as the *non-private* (prompts, responses) of the fresh batch of samples. We use the model $M_{k-1}$ to also provide it's own ranking of these responses (and denote it as $\ell_{M_{k-1}}$). We also have the noisy perturbed labels, $\ell_{RR}$ on the fresh batch of samples. Viewing $(\ell_{RR}, \ell_{M_{k-1}})$ as two noisy observations of the ground truth preference $\ell^*$, we combine these observations and derive the maximum-likelihood estimator (MLE) using both the observations. These MLE estimates act as the labels for this stage of alignment. This process then repeats across $K$ stages. The key insight behind PROPS is that while intermediate LLM models $M_k$ may not yet be fully capable of generating high-quality outputs or responses in the early stages of training, it may still possess sufficient knowledge to correctly label preferences. By utilizing progressively learned models like for preference labeling, PROPS reduces the reliance on noisy perturbed labels, which can degrade the quality of alignment if overused. This staged approach not only improves the quality of the alignment over time but also lessens the burden of maintaining strict privacy guarantees across the entire dataset. Each stage builds upon the knowledge accumulated in previous stages, allowing for more refined preference labeling as the model improves. Thus, our framework leverages the power of intermediate models to enhance alignment efficiency while preserving privacy, offering a novel solution to the challenge of privacy-preserving alignment. We remark here that PROPS can be also used/readily adopted for RLHF based alignment.

- *Comprehensive Experimental Results:* We conduct a comprehensive set of experiments to evaluate the impact of preference-level differential privacy (DP) on alignment performance with different models (namely Pythia-1B, GPT2-Large and GPT2-Medium) across various privacy regimes. Specifically, we focus on DPO-based methods for alignment and analyze how the Win-Tie-Lose rates of privately aligned DPO models vary as a function of the

privacy budget $\epsilon$. First, we compare Randomized Response (RR) based DPO against SFT model to show the benefit of incorporating privacy during alignment. Second, we compare the Win-Tie-Lose rates of our proposed method against conventional DP-SGD that shows the benefit of using preference-level privacy over centralized differential privacy for alignment. In Figure 1, we highlight results demonstrating that in high-privacy regimes, the preference-privacy-based PROPS approach outperforms DP-SGD for two different models: GPT2-Large and GPT2-Medium. We also analyze the effect of model size and privacy constraints when using PROPS compared to RR-DPO, and show that the gains from PROPS are prominent for larger model sizes. We present the Win-Tie-Lose rate for two different datasets (`truthy-dpo-v0.1` and Anthropic HH-RLHF) on GPT2-Medium model which show the consistent improvement provided by PROPS. The detailed experimental results along with discussions are provided in Section 4.

## 2 BACKGROUND ON ALIGNMENT AND PRIVACY

***Model Alignment:*** Alignment of pre-trained LLM models enable the models adhere to human values and societal norms by adjusting the generated responses. Reinforcement Learning based Human Feedback (RLHF) (Stiennon et al., 2020; Bai et al., 2022) and Direct Preference Optimization (DPO) (Rafailov et al., 2024) are two pre-dominant methods for alignment. In this paper, we focus on the DPO-based method for alignment. We start with a preference dataset $\mathcal{D}$ with $n$-samples where the $i^{th}$ sample can be expressed as $(x_i, y_1^i, y_2^i, \ell_i^*)$ where $x_i$ is the prompt, $y_1^i, y_2^i$ are two LLM generated responses and $\ell_i^*$ is the human chosen label that can be defined as:

$$\ell_i^* = \begin{cases} 1 & \text{if } y_1^i \text{ is preferred over the response } y_2^i, \\ 0 & \text{otherwise.} \end{cases}$$

For the ease of notation, we define $y_p$ is the preferred response and $y_{np}$ is the not-preferred response, and suppress the index $i$. Specifically, if $\ell^* = 1$, then we will have $y_p = y_1, y_{np} = y_2$, and for the case $\ell^* = 0, y_p = y_2, y_{np} = y_1$. For a prompt $x$ in the dataset $\mathcal{D}$ with $y_p$ response preferred over the response $y_{np}$, we define the DPO (instance specific) loss as:

$$\begin{aligned} \mathsf{loss}^*(x, y_p \succ y_{np}) &= \log \sigma \left( \beta \log \frac{\pi_\theta(y_p|x)}{\pi_{\mathrm{ref}}(y_p|x)} - \beta \log \frac{\pi_\theta(y_{np}|x)}{\pi_{\mathrm{ref}}(y_{np}|x)} \right) \\ &= \mathbb{1}(\ell^* = 1) \cdot \mathsf{loss}(x, y_1 \succ y_2) + \mathbb{1}(\ell^* = 0) \cdot \mathsf{loss}(x, y_2 \succ y_1). \end{aligned} \quad (1)$$

The instant-specific true loss mentioned in equation 1 represents the loss for every prompt $x$ in preference data $\mathcal{D}$, therefore the expected DPO loss can be defined as:

$$\mathbb{E}[\mathsf{loss}(x, y_1, y_2, \ell^*)] = \mathbb{E}_{(x,y_1,y_2,\ell^*) \sim \mathcal{D}} [\mathbb{1}(\ell^* = 1) \cdot \mathsf{loss}(x, y_1 \succ y_2) + \mathbb{1}(\ell^* = 0) \cdot \mathsf{loss}(x, y_2 \succ y_1)] \quad (2)$$

***Privacy for Alignment:*** The notion of Differential Privacy (DP) Dwork et al. (2014) has been adopted in the alignment frameworks to ensure that the presence or absence of a single sample in a preference dataset does not *significantly* alter the outcome of the model.

**Definition 1** (($\epsilon, \delta$) **Differential Privacy**) *For all pair of neighboring datasets $\mathcal{D}$ and $\mathcal{D}'$ that differ by a single entry, i.e., $||\mathcal{D} - \mathcal{D}'||_1 \leq 1$, a randomized algorithm $\mathcal{M}$ with an input domain of D and output range $\mathcal{R}$ is considered to be $(\epsilon, \delta)$-differentially private, if $\forall \mathcal{S} \subseteq \mathcal{R}$:*

$$\mathbb{P}[\mathcal{M}(\mathcal{D}) \in \mathcal{S}] \leq e^\epsilon \cdot \mathbb{P}[\mathcal{M}(\mathcal{D}') \in \mathcal{S}] + \delta.$$

The DP framework ensures that the inclusion or exclusion of a single data entry $\{x_i, y_1^i, y_2^i, \ell_i^*\}$ does not significantly influence the aligned model's behavior. However, the traditional DP framework may offer a stronger privacy guarantee than what is required since it not only privatize the human generated label $\ell_i^*$ in the context of LLM alignment that can be achieved by exploiting the idea of Label-DP (Chaudhuri & Hsu, 2011; Ghazi et al., 2021b), a relaxed version of DP which specifically aims to protect the privacy of the labels of the dataset, may offer a more ideal solution.

**Definition 2** ($\epsilon, \delta$-**Label DP**) *(Chaudhuri & Hsu, 2011; Ghazi et al., 2021a) For all neighboring datasets $\mathcal{D}$ and $\mathcal{D}'$ that differ by one preference ranking (i.e. $\{x_i, y_1^i, y_2^i, \ell_i\} \in \mathcal{D}$ and $\{x_i, y_1^i, y_i^2, (1 - \ell_i)\} \in \mathcal{D}'$, with privacy parameters $\epsilon \in \mathbb{R}_{>0}$, and $\delta \in [0, 1)$ an algorithm $M$, whose output domain $S$ consists of all possibly aligned models, will satisfy $\epsilon$-preference privacy if :*

$$\mathbb{P}[M(D) \in S] \leq e^\epsilon \cdot \mathbb{P}[M(D') \in S] + \delta. \quad (3)$$

For the case with $\delta = 0$, the model $M$ satisfies $\epsilon$-label DP. This motivates the notion of Preference-level privacy that places a guarantee on labeler's privacy while not significantly impacting a model during its alignment stage. Specifically, preference level privacy ensures that the resulting LLM after alignment should not be significantly impacted by a change in a single preference. Similar to Label-DP, the randomized response mechanism (Warner, 1965) is a direct approach for implementing preference privacy, as the preference $\ell_i$ of a data entry is flipped with probability $\gamma_\epsilon = \frac{1}{1+e^\epsilon}$, where $\gamma_\epsilon$ is used for convenience in notation and $\epsilon$ represents the privacy budget. We define $\epsilon$-preference level DP and the randomized response mechanism as follows:

**Definition 3** (($\epsilon, \delta$)-**Preference level DP**) *For all neighboring datasets $\mathcal{D}$ and $\mathcal{D}'$ that differ by one preference ranking (i.e. $\{x_i, y_1^i, y_2^i, \ell_i\} \in \mathcal{D}$ and $\{x_i, y_1^i, y_i^2, (1-\ell_i)\} \in \mathcal{D}'$, a model after performing an alignment procedure $M$, whose output domain $S$ consists of all possibly aligned models, will satisfy $\epsilon$-preference privacy if :*

$$\mathbb{P}[M(D) \in S] \leq e^\epsilon \cdot \mathbb{P}[M(D') \in S] + \delta. \tag{4}$$

*From Preference-level DP to Labeler-level DP:* Preference-level DP would be a strong guarantee for each labeler if each labeler (user) labels only one prompt's responses. However, in practice this is not the case, and typically each labeler can annotate multiple responses. In this case labeler-level privacy safeguards all instances from the same user (labeler). In this scenario, the notion of adjacency of datasets would be defined w.r.t. presence/absence of a labeler. This distinction has been widely noted and studied in the literature, including including (McMahan et al., 2017; Liu et al., 2020; Levy et al., 2021). One can use techniques from privacy accounting and composition techniques to convert preference-level DP guarantees to labeler-level DP. For the remainder of this paper, we focus on the design of preference-level DP techniques for alignment as the transformation to user level DP can be done by above techniques.

**Randomized Response Based Approach**: As a first baseline solution, one can achieve $\epsilon$-Preference level DP by Randomized Response (RR). For each entry of the preference dataset $\mathcal{D}$, $\{x, y_1, y_2, \ell^*\}$, the output of the RR mechanism is $\{x, y_1, y_2, \ell_{RR}\}$, where:

$$\ell_{RR} = \begin{cases} \ell^* & \text{with probability } (1 - \gamma_\epsilon) = \frac{e^\epsilon}{1+e^\epsilon} \\ 1 - \ell^* & \text{with probability } \gamma_\epsilon = \frac{1}{1+e^\epsilon}. \end{cases} \tag{5}$$

More generally, for a private label generated with RR, $\ell_{RR}$, the private loss on individual prompt $x \in \mathcal{D}$ can be defined as:

$$\mathsf{loss}(x, y_1, y_2, l_{RR}) = \mathbb{1}(l_{RR} = 1) \cdot \mathsf{loss}(x, y_1 \succ y_2) + \mathbb{1}(l_{RR} = 0) \cdot \mathsf{loss}(x, y_2 \succ y_1). \tag{6}$$

From the above equation, the expected DPO loss with randomized response can be written as:

$$\begin{aligned} \mathbb{E}_{\ell_{RR}}[\mathsf{loss}(x, y_1, y_2, \ell_{RR})] =& \mathbb{P}(\ell_{RR} = \ell^*) \cdot [\mathsf{loss}(x, y_1, y_2, l_{RR}|l_{RR} = \ell^*)] + \\ & \mathbb{P}(\ell_{RR} \neq \ell^*) \cdot [\mathsf{loss}(x, y_1, y_2, \ell_{RR}|l_{RR} \neq \ell^*)] \\ =& (1 - \gamma_\epsilon) \cdot \mathsf{loss}[x, y_1, y_2, \ell^*] + \gamma_\epsilon \cdot \mathsf{loss}[x, y_1, y_2, 1 - \ell^*] \end{aligned} \tag{7}$$

Thus, we observe that the DPO loss on the private preference is a biased estimate of the non-private DPO loss. To mitigate this issue, we can instead define a new loss function for DPO training as follows:

$$\mathsf{loss}^{\text{unbiased}}(x, y_1, y_2, \ell_{RR}) = \begin{cases} \frac{1}{1-2\gamma_\epsilon}((1-\gamma_\epsilon)\mathsf{loss}(x, y_1 \succ y_2) - \gamma_\epsilon\mathsf{loss}(x, y_2 \succ y_1)), & \text{if } \ell_{RR} = 1, \\ \frac{1}{1-2\gamma_\epsilon}((1-\gamma_\epsilon)\mathsf{loss}(x, y_2 \succ y_1) - \gamma_\epsilon\mathsf{loss}(x, y_1 \succ y_2)), & \text{if } \ell_{RR} = 0 \end{cases} \tag{8}$$

We note that the above loss function is an unbiased estimate of the true DPO loss (defined w.r.t. the ground truth human preferences), i.e., $\mathbb{E}[\mathsf{loss}^{\text{unbiased}}(x, y_1, y_2, l_{RR})] = \mathsf{loss}(x, y_p \succ y_{np})$ . This result was shown in Ghazi et al. (2021b).

*Remark on Connections to & Differences from robust DPO:* We note that the problem of preference-level privacy is similar to the problem of robust alignment in presence of noisy preferences. Specifically, recent works, including Mitchell (2023), Chowdhury et al. (2024a) and Chowdhury et al. (2024b) study the robustness of alignment when the human-annotated labels are intrinsically noisy. The distinction however, is the following: in our setting, the injected noise (and more importantly

the parameters of the privacy preserving mechanisms (detailed in the next Section)) are known and can be controlled as a function of the privacy parameters. Furthermore, we remark here that some of the theoretical results obtained in Chowdhury et al. (2024a) can be readily applied to our problem and assess the utility-privacy tradeoff of RR based methods. To avoid repetition, we choose not to repeat these kind of results in this paper.

## 3 PROGRESSIVELY PRIVATE SELF-ALIGNMENT (PROPS)

In this section, we present Progressively Private Self-Alignment algorithm (PROPS), which is the main technical contribution of this paper. To facilitate understanding, we first describe PROPS in a two-stage ($K = 2$) setting and the generalization to arbitrary number of stages is straightforward. We begin with the preference dataset $\mathcal{D}$ consists of $n$ samples. Each sample is represented as $(x, y_1, y_2, \ell^*)$, where $x$ is the prompt, $(y_1, y_2)$ are the large language model (LLM) generated responses, and $\ell^*$ is the human labeler's preference. We partition this dataset into two halves, denoted as $\mathcal{D}_1$ and $\mathcal{D}_2$. Let us perturb the labels of each entry using the RR mechanism, i.e., the labels are flipped a probability $\gamma_\epsilon = 1/(1 + e^\epsilon)$.

Stage 1: In the first stage, we use the dataset $D_1$ (with perturbed labels) and use it to align a fine-tuned model via DPO. Let us denote the resulting model as $M_1$. First note that since the training was done on private (perturbed preferences), due to post-processing the model $M_1$ can be used in subsequent stages without additional leakage.

Stage 2: In this stage, we use the dataset $D_2$, and use the model $M_1$ (of the previous stage) to label/rank the preference of each prompt/response-pairs. Note that this procedure only requires the prompt and response pairs (and not the ground-truth human preferences); thus, this does not cause any additional privacy leakage.

Let us denote the corresponding label obtained from the model $M_1$ for a prompt as $\ell_{M_1}$. To summarize, at this point, for each tuple $(x, y_1, y_2)$, we have access to $(\ell_{RR}, \ell_{M_1})$. Viewing these as two *noisy* observations of the ground truth $\ell^*$, it is natural to ask the following: can one combine these predictions to obtain a better prediction about the ground truth preference? Note that we know the error rate of RR (which is $\gamma_\epsilon$), however, we do not know the error rate of the model $M_1$ (say $\gamma_{M_1}$, which denotes the probability with which the model $M_1$ makes errors). However, if we knew the error rate of the model $M_1$ (or an estimate for $\gamma_{M_1}$), then we could then use a combining approach (e.g., the maximum-likelihood esti-

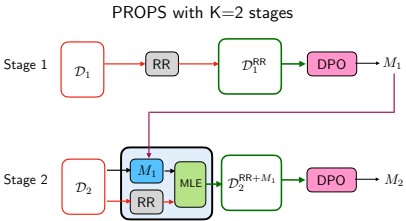

Figure 3: Illustration of PROPS ($K = 2$ stages).

mator or MLE) to design a potentially better estimate of the ground truth label for alignment. In fact, it is not too difficult to work out the MLE combiner using two noisy observations. Assuming that the RR noise and the noise induced by the model $M_1$ are independent, the MLE statistic (log-likelihood ratio) can be written and simplified as follows (see Proof in the Section 6):

$$\Lambda(\ell_{RR}, \ell_{M_1}) = \log\left(\frac{\mathbb{P}(\ell_{RR}, \ell_{M_1} \mid \ell^* = 0)}{\mathbb{P}(\ell_{RR}, \ell_{M_1} \mid \ell^* = 1)}\right) \tag{9}$$

$$= (-1)^{\ell_{RR}} \cdot \log\left(\frac{1 - \gamma_\epsilon}{\gamma_\epsilon}\right) + (-1)^{\ell_{M_1}} \cdot \log\left(\frac{1 - \gamma_{M_1}}{\gamma_{M_1}}\right) \tag{10}$$

The above then yields the methodology one can use for creating a new label for each prompt as:

$$\ell_{\text{PROPS}}(\ell_{RR}, \ell_{M_1}) = \begin{cases} 1 & \text{if } \Lambda(\ell_{RR}, \ell_{M_1}) \leq 0, \\ 0 & \text{if } \Lambda(\ell_{RR}, \ell_{M_1}) > 0. \end{cases} \tag{11}$$

With access to the new label estimates, $\ell_{\text{PROPS}}$ (for all samples in the set $\mathcal{D}_2$), we then train $M_1$ using DPO to obtain a new model $M_2$. This procedure can be repeated in a multi-stage setting by replacing $M_1$ by $M_{k-1}$ which is then trained on PROPS labels to obtain the model $M_k$ for the next stage.

Estimating $\gamma_{M_1}$: Now, the remaining challenge is to estimate $\gamma_{M_1}$, the rate at which the model $M_1$ predicts incorrect labels. We next present an interesting approach to estimate $\gamma_{M_1}$. Suppose that

we make the hypothesis that the model $M_1$ also independently flips the ground truth labels with probability $\gamma_{M_1}$, then we can write the output of the model and the output of the RR mechanisms as follows:

$$\ell_{RR} = \ell^* \oplus U, \quad \ell_{M_1} = \ell^* \oplus V, \tag{12}$$

where $U \sim \text{Bernoulli}(\gamma_\epsilon)$ and $V \sim \text{Bernoulli}(\gamma_{M_1})$. Thus, estimating $\gamma_{M_1}$ is equivalent to estimating the parameter of the Bernoulli random variable $V$. Note that we have $|\mathcal{D}_2|$ such observations (one for each sample in the second half of the dataset). If we compute

$$\mu_{M_1} = \frac{\sum_i \ell_{M_1}^{(i)} \oplus \ell_{RR}^{(i)}}{|\mathcal{D}_2|}, \tag{13}$$

which represents the number of disagreements between the labels predicted by RR and the model $M_1$, this value in fact converges to the the expected value of $\mathbb{E}[U \oplus V] = \gamma_{M_1}(1-\gamma_\epsilon) + \gamma_\epsilon(1-\gamma_{M_1})$. Since we know $\gamma_\epsilon$, we can then use it to compute the unknown parameter $\gamma_{M_1}$. This leads us to propose the following estimator for $\gamma_{M_1}$:

$$(\text{Estimate of } \gamma_{M_1}) \quad \hat{\gamma}_{M_1} = \frac{\mu_{M_1} - \gamma_\epsilon}{1 - 2\gamma_\epsilon}. \tag{14}$$

The detailed proof of the above result and the fact that above estimator is unbiased is presented in the Section 6. With all the above, we have all the ingredients for PROPS.

---

**Algorithm 1** PROgressively Private Self-alignment (PROPS)

---

**Inputs:** Fine-tuned Model $M_0$, Dataset $\mathcal{D}$, Stages $K$, Privacy parameters $(\epsilon, \delta)$
**Output:** Aligned model $M_K$

Perform RR on dataset $\mathcal{D}$, such that $\mathcal{D} \xrightarrow{RR(\gamma_\epsilon)} \mathcal{D}'$
Partition the dataset $\mathcal{D}'$ into $K$-disjoint datasets such that $\mathcal{D}' = D_1 \cup D_2 \cup \ldots \cup D_K$
Flip labels using Randomized Response (RR) with probability $\gamma_\epsilon = \frac{1}{1+e^\epsilon}$
Align model $M_0$ with dataset $D_1$ to get model $M_1$
**for** $k = 2, 3, \ldots, K$ **do**
    Generate labels $\ell_{M_{k-1}}^{D_k}$ for dataset $D_k$ using the model $M_{k-1}$ (from previous stage)
    Obtain $\ell_{RR}^{D_k}$ and $\ell_{M_{k-1}}^{D_k}$ and obtain the maximum likelihood estimator (MLE) $\Lambda$ according to equation 15 and generate label as:

$$\ell_{\text{PROPS}}^{D_k} = \begin{cases} 1 & \text{If } \Lambda \leq 0, \\ 0 & \text{If } \Lambda > 0. \end{cases}$$

    Align model $M_{k-1}$ on dataset $D_k$ with PROPS labels ($\ell_{\text{PROPS}}^{D_k}$) to get model $M_k$
**end for**
**Output:** Aligned model $M_K$ on dataset $\mathcal{D}$

---

***PROPS Algorithm & Remarks*** We present the main algorithm of this paper (PROgressively Private Self-alignment) PROPS in Algorithm 1. We now make a sequence of remarks regarding this algorithm: (a) *Privacy guarantees of PROPS:* PROPS satisfies $(\epsilon, 0)$-preference-level differential privacy (DP). Furthermore, techniques such as sub-sampling can be readily combined with our idea for privacy amplification. The key aspect of PROPS is that even though it does not explicitly reduce privacy leakage, for the same leakage as one would achieve by a *vanilla* RR mechanism, it can potentially provide higher utility (as PROPS trains on potentially less noisy preferences compared to vanilla RR). (b) *Compatibility with RLHF and other alignment approaches:* While we have presented the idea with DPO in the backdrop, the exact same procedure is applicable for alignment using RLHF based algorithms (as these algorithms also require labeled prompt-response pairs). (c) The description of the algorithm satisfies $(\epsilon, 0)$-preference level DP. However, we can *lift* the algorithm to achieve user/labeler level privacy as discussed in Remark in Section 2. Specifically, achieving labeler level DP would require a bound on the maximum number of responses (per labeler), which can then be used together with composition techniques to provide labeler-level DP guarantees. (d) *Distinction from Multi-Stage RR:* We would also like to note that even though PROPS bears similarities to Multi-Stage RR Ghazi et al. (2021b), it has important differences. In multi-stage RR, in

each stage the labels for the next stage are obtained by either using RR or via the model from the previous stage (e.g., using a sampling strategy). The distinction lies in the fact how one combines the noisy labels and the model predictions. Instead of using a simple approach of randomly selecting between the two, we instead first estimate the error rate of the model, which are then combined using an MLE approach. Secondly, we note that in LLM alignment, the models are not directly trained for preference classification, but rather for responding to prompts in a socially acceptable manner. The versatility of LLMs and their ability to handle multiple tasks allows us to use the LLMs for dual purposes; namely preference classification for self-alignment. (e) *Distinction between PATE and PROPS:* We highlight key distinctions between PATE Papernot et al. (2018) and PROPS. First, PATE provides the privacy of all features; whereas in this paper we focused on preference-level-privacy. Second, PROPS is based on a sequential and iterative approach. In each iteration (stage), we use the model learned from the previous stage to rank the prompt/responses and combine them with the RR-perturbed rankings. These combined rankings (which are obtained using a Maximum-Likelihood-combiner) are then used to train the model in the next iteration using DPO. On the other hand, PATE follows a parallel-training paradigm; namely it trains multiple teacher models on data partitions in parallel, whose outputs (labels) are then aggregated with DP. These labels are then used for subsequent training. (f) *Impact of number of stages and privacy budget in PROPS:* The number of stages in the PROPS framework can be treated as a hyperparameter and optimized based on the dataset. Since the 2-stage PROPS approach was already outperforming baseline methods, we did not extend our comparisons to multi-stage settings in our experiments. However, a carefully chosen number of stages has the potential to further enhance the framework's privacy-utility tradeoff.

## 4 EXPERIMENTAL SETUP AND RESULTS

In this Section, we present results on $\epsilon$-preference level DP for DPO, and compare the results with our proposed algorithm PROPS. The code for PROPS along with the used datasets are publicly available . In our experiments and validation, we have used (1) two datasets (`jondurbin/truthy-dpo-v0.1` dataset and Anthropic HH-RLHF ) and (2) three different models of varying sizes: Pythia-1B EleutherAI (2024) , GPT2-Large (774M) OpenAI (2024a) and GPT2-Medium (355M) OpenAI (2024b). We have adopted a similar experimental setup as the prior works Rafailov et al. (2024); Chakraborty et al. (2024); von Werra et al. (2020)Lior-Baruch (2024) Our results are organized as follows: (a) We first show how RR-based scheme and our method (PROPS) impact the Win-Tie-Lose rate relative to a SFT model, and the improvement provided by PROPS compared to RR. (b) We also compare PROPS vs DP-SGD for the three models and the two datasets to show the consistent advantage of PROPS.

**Impact of Preference Privacy on Win-rate Performance**. We now present results using the PROPS algorithm. We implemented a two-stage ($K = 2$) procedure, where the first stage aligns the initial SFT model using noisy RR perturbed labels to obtain a model $M_1$; in the second stage, we use model $M_1$'s predictions together with noisy labels (of the second half of the dataset) to obtain new PROPS labels. We then train the model $M_1$ using PROPS labels to obtain the final model $M_2$. Figure 4 reports the win-rate for the privately trained DPO models with respect to the fine-tuned model (denoted as SFT for supervised fine-tuning) used before DPO assuming the `truthy-dpo-v0.1` dataset and `GPT-2-Large` model.

The objective of this experiment is to observe if the privately trained DPO models can still attain a comparable utility while enforcing a stronger privacy guarantee. Specifically, Figure 4 shows two sets of win-tie-loss rate results: PROPS against SFT and RR against SFT. We report results in the practically relevant high-privacy regime, evaluating Label-DP for four values of $\epsilon$: 0.1, 0.5 1, 2. The results indicate that the privately trained methods can still generate responses aligned with societal val-

| Privacy Budget | Win-Tie-Lose rate | |
|---|---|---|
| | RR vs SFT | PROPS vs SFT |
| $\epsilon = 0.1$ | $48 : 3 : \mathbf{49}$ | $41 : 6 : \mathbf{53}$ |
| $\epsilon = 0.5$ | $\mathbf{54} : 3 : 43$ | $\mathbf{67} : 3 : 30$ |
| $\epsilon = 1$ | $\mathbf{54} : 9 : 37$ | $\mathbf{59} : 0 : 41$ |
| $\epsilon = 2$ | $\mathbf{71} : 2 : 27$ | $\mathbf{56} : 7 : 37$ |

Figure 4: Relative performance of RR and PROPS based alignment versus SFT model.

ues compared to their original SFT counterparts while guaranteeing preference DP. As expected, for stricter privacy budgets (i.e. $\epsilon = 0.1, 0.5$) the win-rate is smaller compared to more relaxed privacy budgets (i.e. $\epsilon = 1, 2$).

**PROPS vs RR:** We report results comparing the responses of PROPS vs. RR mechanism in Figure 5 in terms of Win-Tie-Lose rate. We used GPT-4 as the evaluator and `truthy-dpo-v0.1` dataset was used for alignment. To analyze the impact of model size, we ran this comparison for `Pythia-1B`, `GPT2-Large`, and `GPT2-Medium` models on the `truthy-dpo-v0.1` dataset

comparing our PROPS method with randomized response (RR) in Figure 5. The results indicate that PROPS can significantly outperform RR when using a larger model such as `Pythia-1B`. Furthermore, PROPS can benefit larger models more consistently (i.e. across varying privacy budgets) more than RR. For `Pythia-1B`, PROPS beats RR at all privacy budgets and for `GPT2-Large` (774 million parameters) beats RR at all budgets except $\epsilon = 0.5$, where it loses by a slight margin. For `GPT2-Medium` (355 million parameters), an interesting trend is observed where PROPS beats RR at $\epsilon = 0.1, 2$ but loses at $\epsilon = 0.5, 1$. This could be a result of `GPT2-Medium` having fewer parameters than `GPT2-Large` and `Pythia`, as it would have lower generalization capabilities causing it to make incorrect predictions during the subsequent stages of PROPS.

| Privacy Budget | PROPS vs RR (Win-Tie-Lose) | | |
|---|---|---|---|
| | Pythia (1B) | GPT2-Large | GPT2-Medium |
| $\epsilon = 0.1$ | **68** : 4 : 28 | **62** : 2 : 36 | **68** : 8 : 24 |
| $\epsilon = 0.5$ | **64** : 3 : 33 | 44 : 10 : **46** | 44 : 7 : **49** |
| $\epsilon = 1$ | **57** : 5 : 38 | **54** : 11 : 35 | 42 : 9 : **49** |
| $\epsilon = 2$ | **51** : 11 : 38 | **49** : 28 : 23 | **49** : 6 : 45 |

Figure 5: Comparison of Win-Tie-Lose rate for PROPS vs RR for high privacy regimes with three models: Pythia, GPT2-Large, GPT2-Medium.

**PROPS vs DPSGD:** In Figure 6, we present Win-Tie-Lose rates comparing our proposed algorithm PROPS with the conventional DP-SGD algorithm for `GPT2-Large` and `GPT2-Medium` models on the `truthy-dpo-v0.1` dataset. DP-SGD was ran using the Gaussian mechanism for 1 epoch with $\delta = 10^{-10}$, a gradient clipping threshold of 10, and a batch size of 4. PROPS was ran for 2 epochs with a batch size of 4. As the results indicate, PROPS is able to consistently outperform DP-SGD at higher privacy regimes ($\epsilon = 0.1, 0.5, 1$) for both models. This indicates that while DP-SGD attempts to additionally protect the prompts and responses, it suffers a significant drop in utility for smaller privacy budgets. One critical distinction to highlight is that the implementation of PROPS ensures a pure differential privacy (DP) guarantee, specifically $(\epsilon, 0)$-DP. In contrast, DP-SGD provides an approximate differential privacy guarantee, denoted as $(\epsilon, \delta)$-DP. Next, we present results to show the mean and standard deviation of the normalized Win-Tie-Lose rates for PROPS vs DP-SGD on (a) HH-RLHF dataset and (b) `truthy-dpo-v0.1` datasets in Figure 7 for four privacy parameters. The table indicates that PROPS on-average outperforms DP-SGD on both datasets at high privacy regimes (details regarding the experiments are presented in the Appendix).

| Privacy Budget | PROPS vs DP-SGD (Win-Tie-Lose) | |
|---|---|---|
| | GPT2-Large | GPT2-Medium |
| $\epsilon = 0.1$ | **66** : 2 : 32 | **76** : 1 : 23 |
| $\epsilon = 0.5$ | **60** : 2 : 38 | **50** : 11 : 39 |
| $\epsilon = 1$ | **60** : 3 : 37 | **49** : 7 : 44 |
| $\epsilon = 2$ | 44 : 6 : **50** | **54** : 4 : 42 |

Figure 6: Comparison of Win-Tie-Lose rate for PROPS vs DP-SGD for high privacy regimes with GPT2-Large, GPT2-Medium models.

| Privacy Budget | PROPS vs DP-SGD (Win-Tie-Lose) | |
|---|---|---|
| | truthy-dpo-v0.1 dataset | HH-RLHF dataset |
| $\epsilon = 0.1$ | $77.6 \pm 2.19 : 2 \pm 2 : 20.4 \pm 0.89$ | $50 \pm 8.24 : 19.2 \pm 7.29 : 30.8 \pm 7.15$ |
| $\epsilon = 0.5$ | $53.6 \pm 6.69 : 8.4 \pm 1.6 : 38 \pm 7.07$ | $46.8 \pm 6.41 : 27.2 \pm 7.01 : 26 \pm 2.44$ |
| $\epsilon = 1$ | $52 \pm 6.78 : 4.8 \pm 4.14 : 43.2 \pm 5.93$ | $54.4 \pm 9.52 : 18 \pm 6.78 : 27.6 \pm 7.92$ |
| $\epsilon = 2$ | $65.6 \pm 4.34 : 2 \pm 2 : 32.4 \pm 3.28$ | $32.8 \pm 5.4 : 25.2 \pm 7.29 : 42 \pm 7.87$ |

Figure 7: Mean and Standard Deviation for 5 independent runs on `truthy-dpo-v0.1`, and HH-RLHF datasets for high-privacy and moderate-privacy regimes with GPT2-Medium model.

**Illustrative Example Responses to Prompts for varying privacy levels**: To further illustrate the privacy-utility tradeoff, we analyze the responses of LLMs trained at different privacy budgets for the same prompt. Figure 8 presents responses for models trained when $\epsilon = 0, 1, \infty$ for two different prompts. For instance, the first prompt is: "I've been seeing a lot of slugs outside recently, even crawling up trees. Should I do something about them, or just let them be?". As the table indicates, the model trained with $\epsilon = 0$ (corresponding to random flipping) only regurgitates the question to the user. The model trained with $\epsilon = 1$ does produce an actual response but introduces its own opinion/bias on the matter. The model trained with $\epsilon = \infty$ gives a more "professional" answer but is not very helpful. For the second prompt, an interesting trend can be observed where the models trained when $\epsilon = 0, \infty$ behave similarly to the previous example, but at $\epsilon = 1$ a more informative answer is given. This again indicates that privacy can be ensured while still attaining a decent utility. It should be noted that quality assessment of language models is inherently subjective and could alternatively be analyzed along three individual axes: (a) Coherence, (b) Helpfulness, and (c) Harmlessness. While GPT-4 can provide fine-grained rankings for individual attributes, combining these into an overall judgment remains subjective. When generating the win-tie-loss results in the

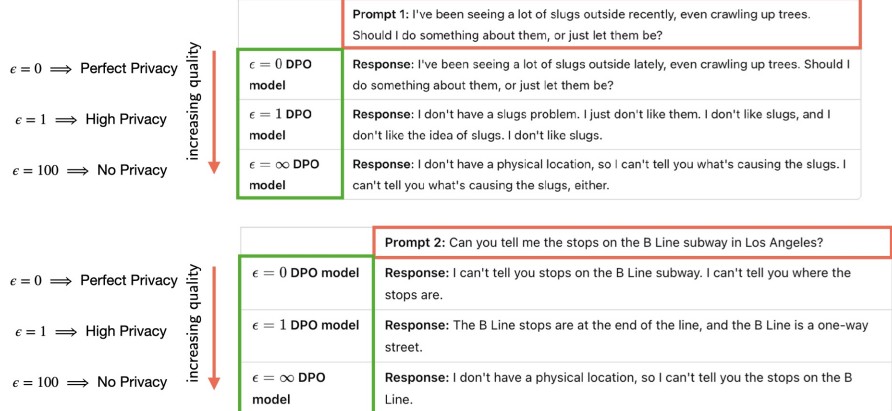

Figure 8: This figure shows examples of responses to two prompts (denoted as Prompts 1 & 2) for three different aligned models using $\epsilon$-Preference-level DP: $\epsilon = 0$ DPO model was trained on completely randomly labeled preferences; $\epsilon = \infty$ DPO model refers to the non-privately aligned model. We can clearly observe a perceptual increase in response quality; moreover, the response remains coherent even at a high privacy regime (corresponding to $\epsilon = 1$).

| Privacy Budget | Response | Coherence | Helpfulness | Harmlessness |
|---|---|---|---|---|
| $\epsilon = 0.1$ | Yes, it does. But it also protects the speech of others. The First Amendment protects the speech of others, but it also protects the speech of the speaker. The First Amendment. | 4 | 3 | 10 |
| $\epsilon = \infty$ | This is a misconception due to the common belief that the First Amendment protects individuals from restrictions on speech. | 6 | 4 | 10 |

Figure 9: Comparing individual ratings of coherence, helpfulness, and harmfulness of GPT-4 when given prompt: "Does the First Amendment protect individuals from restrictions on speech imposed by private businesses and individuals?" for GPT2-Large trained with PROPS at $\epsilon = 0.1$ and when $\epsilon = \infty$ (i.e. non-private).

main paper, we focused on prompting GPT-4 to equally prioritize coherence, helpfulness and harmfulness when evaluating responses. We did this so that we could assess the win-tie-lose statistics for each model across a large sample size of prompts. Having said that, if we were to assess each response individually one can in principle ask GPT-4 to provide fine-grained rankings for each attribute (coherence, helpfulness, harmlessness). However, combining these rankings to rate or judge a response as superior becomes subjective. As an example we show these fine-grained rankings for two responses from the same prompt at different privacy budgets in Figure 9 under varying privacy budgets for GPT2-Large.

## 5 CONCLUSIONS

In this paper, we presented new results towards aligning LLMs with preference level privacy, which preserve the privacy of preferences provided by humans. We build and expand upon the concept of label DP for this problem, and present a series of increasingly sophisticated, yet practical privacy preserving mechanisms for alignment. Specifically, starting from a standard randomized response (RR) mechanism which randomly flips human preferences, we presented a new mechanism, PROPS (PROgressively Private Self-alignment) which works across multiple stages. The key insight behind PROPS is that while intermediate LLM models may not yet be fully capable of generating high-quality outputs or responses in the early stages of training, it may still possess sufficient knowledge to correctly label preferences. Thus, our framework leverages the power of intermediate models to enhance alignment efficiency while preserving privacy, offering a novel solution to the challenge of privacy-preserving alignment. We also provided a comprehensive set of experiments on two datasets and multiple model sizes which show that PROPS outperforms DP-SGD and randomized response (RR) based approaches. We quantified and measure these gains in terms of win-tie-lose rates, and these gains are especially substantial in practically relevant high privacy regimes.

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

# 6 APPENDIX

## 6.1 MLE ESTIMATOR FOR $\ell^*$ USING $(\ell_{RR}, \ell_{M_1})$: EQUATION 10

Given the flipped labels $\ell_{RR}$ and $\ell_{M_1}$ by Randomized Response (RR) and model $M_1$ respectively, we aim to come up with a good decision making policy for the proposed algorithm. We calculate the likelihood of observing $(\ell_{RR}, \ell_{M_1})$ given the possible values of $\ell^*$. We define $\gamma_\epsilon$ as the flipping probability of RR and $\gamma_{M_1}$ as the flipping probability of the model.

| $\ell^*$ | $\mathbb{P}(\ell_{RR}=0|\ell^*)$ | $\mathbb{P}(\ell_{RR}=1|\ell^*)$ | $\mathbb{P}(\ell_M=0|\ell^*)$ | $\mathbb{P}(\ell_M=1|\ell^*)$ |
|---|---|---|---|---|
| 0 | $1-\gamma_\epsilon$ | $\gamma_\epsilon$ | $1-\gamma_M$ | $\gamma_M$ |
| 1 | $\gamma_\epsilon$ | $1-\gamma_\epsilon$ | $\gamma_M$ | $1-\gamma_M$ |

Figure 10: The table represents the probability of observing $\ell_{RR}$ and $\ell_M$ based on the flipping probabilities $\gamma_\epsilon$ and $\gamma_{M_1}$ and true label $\ell^*$.

For binary $\ell^*$, now we present the probability of observing specific values of $\ell_{RR}$ and $\ell_{M_1}$. To find the best estimator, we compute the log-likelihood ratio:

$$\Lambda = \log\left(\frac{\mathbb{P}(\ell_{RR}, \ell_{M_1} \mid \ell^*=0)}{\mathbb{P}(\ell_{RR}, \ell_{M_1} \mid \ell^*=1)}\right) \tag{15}$$

$$\overset{(a)}{=} \log\left(\frac{\mathbb{P}(\ell_{RR}|\ell^*=0) \cdot \mathbb{P}(\ell_{M_1}|\ell^*=0)}{\mathbb{P}(\ell_{RR}|\ell^*=1) \cdot \mathbb{P}(\ell_{M_1}|\ell^*=1)}\right)$$

$$\overset{(b)}{=} (-1)^{\ell_{RR}} \log\left(\frac{1-\gamma_\epsilon}{\gamma_\epsilon}\right) + (-1)^{\ell_{M_1}} \log\left(\frac{1-\gamma_{M_1}}{\gamma_M}\right)$$

where, (a) is obtained since $\ell_{RR}$ and $\ell_{M_1}$ are independent, and (b) values are obtained from Table 10. This concludes the proof of equation 10.

## 6.2 ESTIMATOR FOR $\gamma_{M_1}$: PROOF OF EQUATION 14

We have noisy labels $\ell_{RR}$ generated by RR with a flipping probability of $\gamma_\epsilon$, and predicted labels $\ell_{M_1}$ by the model $M_1$ with a flipping (error) probability of $\gamma_{M_1}$. We define $\ell_{RR}$ and $\ell_{M_1}$ as:

$$\ell_{RR} = \ell^* \oplus U, \quad \ell_{M_1} = \ell^* \oplus V, \tag{16}$$

where $U \sim \text{Bernoulli}(\gamma_\epsilon)$ and $V \sim \text{Bernoulli}(\gamma_{M_1})$. We first make the observation that for $i^{th}$ sample in the dataset, $\ell_{M_1}^{(i)} \oplus \ell_{RR}^{(i)} = (\ell_i^* \oplus V_i) \oplus (\ell_i^* \oplus V_i) = V_i \oplus U_i$, where, $U_i$ and $V_i$ are independent. Now, define

$$\mu_{M_1} = \frac{\sum_i \ell_{M_1}^{(i)} \oplus \ell_{RR}^{(i)}}{|\mathcal{D}_2|}$$

and note that $\mu_{M_1}$ is an unbiased estimator for the expected value of $\mathbb{E}[U \oplus V] = \gamma_{M_1}(1-\gamma_\epsilon) + \gamma_\epsilon(1-\gamma_{M_1})$. Hence, we can use $\mu_M$ to obtain an estimate for $\hat{\gamma}_{M_1}$ as follows:

$$\hat{\gamma}_{M_1} = \frac{(\mu_{M_1} - \gamma_\epsilon)}{1 - 2\gamma_\epsilon} \tag{17}$$

This concludes the proof of equation 14.

## 6.3 TRAINING DETAILS

We now summarize how the models were trained for the experiments. The training procedures for each dataset were as follows:

truthy-dpo-v0.1: 15% of the data was used for SFT. 75% of the data was used for passing the model through DPO. This segment of data was split in half, and two epochs of DPO were ran over each half. We then filtered the dataset to keep the preference pairs that were generated by prompting an LLM to be "an honest and helpful assistant" and performed DPO using half of the dataset for 3 epochs. The Win-Tie-Loss rates are calculated using the remaining 10% of the truthy-dpo-v0.1 dataset, which is equivalent to 100 prompts.

HH-RLHF: We use an SFT model[1] available on huggingface that was trained for 1 epoch over the Anthropic-HH dataset. 1000 samples from the test set were used to run DPO. Specifically, these samples were split into 2 halves, and DPO was run for three epochs on each half. While the same prompts were run for both PROPS and DP-SGD, PROPS was fed prompts from the same dataset but with a different format[2]. 100 samples from the same test set (see Footnote 2) were used to generate the Win-Tie-Loss results.

---

[1]https://huggingface.co/jtatman/gpt2-open-instruct-v1-Anthropic-hh-rlhf
[2]https://huggingface.co/datasets/psyche/anthropic-hh-rlhf

