# OpenReview forum: "Aligning Large Language Models With Preference Privacy"
_ICLR.cc/2025/Conference — Submitted to ICLR 2025_

### Official Review · Reviewer_4tSw · 2024-10-25

**Soundness:** 3
**Presentation:** 3
**Contribution:** 2
**Rating:** 5
**Confidence:** 4

**Summary:**

This paper studies alignment in LLMs, focusing on using differential privacy (DP) to protect the privacy of human labeled data. Previous works used full DP for the training, which is superfluous when only the preferences of the labelers need to be protected, rather than the prompt and response pairs. The authors propose improvements by considering randomized response (RR) for Label DP, and also present a new algorithm PROPS for doing a multi-staged training with the noised labels. They run experiments and show that their new methods have good utility compared to nonprivate baselines even at high privacy regimes.

**Strengths:**

- The authors adapt existing techniques to a new setting for aligning LLMs, and propose a new multistage mechanism (PROPS) for training the model.
- PROPS improves over RR for some settings of parameters.

**Weaknesses:**

- The technique in Lemma 1 for unbiasing the loss function is not new (see e.g. the paper “Deep Learning with Label Differential Privacy” by Ghazi et al., in NeurIPS 2021).
- It would be great if the authors included more results about Labeler-level DP, since it is more relevant for applications and it is unclear how much the model utility will be affected.
- The experimental results were only on a single dataset, and it would be good to have results for multiple types of datasets.
- The authors propose their new method as an improvement over existing works with full DP, but do not compare against it in their experiments.

**Questions:**

- How does the number of stages in PROPS affect the utility?
- Why does PROPS perform worse than RR at $\epsilon=0.1$ in Figure 3? Is there a standard deviation for the results?
- Does it improve utility to run PROPS for multiple epochs?

---

### Official Review · Reviewer_nD7n · 2024-11-02

**Soundness:** 3
**Presentation:** 3
**Contribution:** 2
**Rating:** 6
**Confidence:** 3

**Summary:**

The paper studies the problem of aligning large language models with preference level DP, where the binary human preferences for fixed (prompt, response) pairs are the only sensitive information.  The authors propose a randomized-response style mechanisms and variants thereof to solve this problem.  Experimentally, they show how the utility of their mechanism improves with the privacy budget, and that their more sophisticated PROPS algorithm outperforms a simple randomized response approach.

**Strengths:**

1. Principled approach to a timely problem.
2. Demonstrated better-than-baseline performance in one experimental setting.
3. Well written and mostly easy to follow.
4. The illustrative example in Figure 5 is pretty nice.

**Weaknesses:**

1. Experiments are pretty thin and could benefit from more thorough treatment.  Are there any baselines or other prior work you could compare against (e.g., mechanisms for label-DP)?
2. The problem considered is fairly narrow.  Is the core algorithm you proposed more broadly applicable to other settings?  How does your work compare algorithmically to other prior work for label-DP?
3. Technical contribution is fairly small, although I think the simplicity is a good thing here.  But related to (1) above simple approaches would benefit from a more thorough empirical treatment.

**Questions:**

1. In experiments, can you compare against optimizing the biased loss function?
2. RR provides local DP, are there any benefits to only preserving central DP?  I'm not quite sure what that mechanism would look like in this setting, but maybe a discussion somewhere in the paper could be good.

---

### Official Review · Reviewer_Kaiv · 2024-11-04

**Soundness:** 3
**Presentation:** 3
**Contribution:** 3
**Rating:** 5
**Confidence:** 4

**Summary:**

This paper proposed the notion of preference privacy for LLM alignment as an alternative to using DPSGD to protect the privacy of the annotators. Their motivation stems from the fact that DPSGD-based approaches can be excessive, providing more privacy than necessary, and can degrade model utility. Their contributions are in two folds, leveraging DPO as the base alignment algorithm; first, they proposed an unbiased loss for DPO for randomized response method. Secondly, they proposed PROPS, a multi-stage alignment algorithm with preference privacy. In their experiments, they showed that preference privacy based alignment can attain preference a comparable utility to their non-privately aligned counterparts.

**Strengths:**

The authors made two major contribution; the introduction of unbiased estimation loss to DPO based on randomized response and PROPS.
PROPS is a multi-stage algorithm which trains models to generate labels / preferences for the next batch. The preferences and the noisy groundtruth are combined with MLE and are used for the alignment.
I believe that the problem of the privacy risks of alignment is important, and this paper does a fine job in showing how important the problem is.

**Weaknesses:**

The authors did a good job in highlighting the importance of the problem. However, there are a number of claims in the paper that are not experimentally grounded. For instance, one of the major claim of the paper is that DPSGD-based method hurts utility. I don't see such experiments in comparing DPSGD-based method with their proposed preference privacy.
Secondly, a claim is that PROPS can be used for RLHF-based alignment. No experiment in this direction either. While there is a "PROPS Algorithm & Remarks" section, it does not do justice to actually varying the claims.
Lastly, only one LLM (GPT-2 Large) was used for their experiment

**Questions:**

I have the following questions for the authors. I am willing to increase my score if the authors can address my concerns:

- To verify one major claim of the paper, could you please show experimentally that DPSGD indeed hurts privacy compared to the proposed preference privacy? One way would be to compare the DPSGD-based method with the preference privacy under the same privacy budget

- While the proposed PROPS is clever, I see a very high similarity to the idea from PATE [a], where models are trained on disjoint datasets and a noisy label is obtained from the aggregation of the teacher ensembles in a DP fashion? Can you comment on why PATE approach cannot be adapted here?
In fact, I would rather say that PROPS inherently follows the private-public data setup of PATE. Here, what is public is the (prompt, responses) while the label is private.

- Can the authors comment on the need / rationale for using fresh batch of samples for PROPS?

- One major problem is in the experimental setting. While the proof of concept is appreciated, using other language models like the Pythia suite of models [b], Gemma [c], or Llama3 [d] that are current used in practice and showing how different models behave with PROPS alignment would have been a better approach.

- A problem of DPO which PROPS use as a base method is in its generalization to out of distribution. Now that RR is added, how does it affect the performance of PROPS?

- Can the authors provide an experiment on using different alignment like the RLHF. It would make the paper stronger to see such transferability and if there are any potential issues with those alignment approaches.

- In figure 4, why is the win-tie-loss rate of $\epsilon=\infty$ lower than $\epsilon=2$?

- In figure 5, what I can deduce is that training with no privacy ($epsilon=\infty$) leads to somewhat arbitrary response while training with privacy ($\epsilon=1$) can give some reasonable response (Prompt 2). I find this somewhat interesting. To better understand what is going on, can the authors also present the result of DPO without any privacy. Just normal DPO.

- While the motivation sounds plausible, I don't see a real threat here. I understand why DP could be used as a blanket covering for the privacy of the annotators, but the motivation for an annotator being exposed for their preference is somewhat unrealistic to me. Can the authors point to a reference in which this is practical?

[a] Scalable Private Learning with PATE https://arxiv.org/abs/1802.08908

[b] Pythia: A suite for analyzing large language models across training and scaling. https://arxiv.org/abs/2304.01373

[c] Gemma: Open models based on gemini research and technology. https://arxiv.org/abs/2403.08295

[d] llama 3 herd of models. https://arxiv.org/abs/2407.21783

---

> ### Comment · Reviewer_Kaiv · 2024-11-27
> **Thanks for your response**
>
> Dear authors,
>
> Thank you for clarifying my concerns. I'm fairly satisfied with your response.
>
> However, I don't see any of my concerns being incorporated into the paper. I am willing to increase my score if the authors can update their manuscripts. Please color-code the changes for ease of visualization. You can include the additional experiments and clarifications provided in response to my (or any other reviewers') concerns in the Appendix if there is no sufficient space in the main paper.
>
> Thanks again for your work

---

> ### Comment · Reviewer_Kaiv · 2024-11-27
> **Thanks for updating the paper**
>
> Dear authors,
>
> Thanks for updating the paper. I have increased my score

---

### Official Review · Reviewer_Xru6 · 2024-11-06

**Soundness:** 2
**Presentation:** 2
**Contribution:** 2
**Rating:** 5
**Confidence:** 2

**Summary:**

This work addresses privacy in aligning large language models (LLMs) with human values by protecting the privacy of human preferences in feedback data, using a refined approach beyond traditional DP-SGD methods. It introduces PROPS, a mechanism that progressively privatizes model alignment by combining randomized response with partial self-alignment over multiple stages. This method aims to protect preference-level privacy with minimal degradation to model utility, showing promise for reducing reliance on human-labeled samples while maintaining alignment efficacy.

**Strengths:**

Work seems interesting and seems to focus on a problem that is largely overlooked. Algorithm is novel, and in the setting appears to do well. See Questions.

**Weaknesses:**

I remain unconvinced of the prevalence of the label-privacy setting. Further, the method itself is presented well, but I feel that more understanding of how the parameters are selected is needed. Lastly, I’d like a more thorough evaluation. See Questions.

**Questions:**

* You write “We note that in typical alignment scenarios, the prompts and responses are not generated by humans (the two responses (y1 , y2 ) are usually obtained from the fine-tuned LLM)” Why are human preferences only on the labels, and not associated with the prompts? Since humans often are generating the prompts. What setting are humans only voting on prompt-responses? For example, knowing that someone is voting on “what to do if someone has disease A” is revealing.

* I think you should consider looking to PATE, where you have a private dataset that you train with ML, and then ensembling the labels on a public dataset with DP. The setting is slightly different, as you have different labelers label the same data (privately), but it’s quite related, as the labels are private, but the data is public.

* While not as relevant, I’ll point you to PMiXeD (https://arxiv.org/abs/2403.15638) which applies the PATE framework to the predictions of LLMs.

* I’d like more examples of label-privacy machine learning methods in your related works.

* Regarding going from preference-level to labeler-level DP, you should consider looking at research on personalized DP, such as: Individualized PATE (https://arxiv.org/abs/2202.10517). Specifically in Individualized PATE, different teachers can produce different numbers of labels because they have different privacy budgets. Other related works are Individual Privacy Accounting for DPSGD (https://arxiv.org/abs/2206.02617), and (less relevant) Individualized DPSGD (https://arxiv.org/abs/2303.17046)

* Can you make the computational cost of this method more explicit?

* How do you pick the number of rounds you wish to do, and how to split your privacy budget?

**Evaluation:**
* Why is this win-tie-loss rate for this specific dataset the right measure here?
*  It’s hard to tell what the non private baseline is, and the existing baseline
* You seem to focus on “quality” but it seems like you have 2 axes that you care about: quality and helpfulness/relevance. It does not seem like these two measures should be combined into 1 (e.g. “i don’t have a physical location” is given as a high-quality answer, but it is clearly not helpful, as you state yourself). How should we interpret the performance of your model on these two axes?
* For figure 5, it’s hard to tell if your results in the figure are repreesntative, or just those 2 examples.


**Typo:**
Abstract uses “it’s” not “its” - “mechanism which randomly flips human preferences, and it’s corresponding unbiased”

---

### Meta-Review · Area_Chair_Hdp2 · 2024-12-14

**Metareview:**

## Summary of Contributions

This paper studies the problem of LLM alignment with human feedback. The concern here is that these feedback might leak sensitive information of the labeler. The authors propose to use (label) differential privacy to tackle this problem. They propose two algorithms: randomized response (RR) and PROPS (PROgressively Private Self-alignment). For the former, the human's label is randomly flipped to achieve DP; this is used in conjunction with an appropriately-debiased loss function. For the latter, in addition to these flipped label, the prediction of the model trained so far is also used to infer the label.

## Strengths

- LLM alignment is an important process, and privacy protect indeed has not been sufficiently explored.
- The proposed algorithm (PROPS) performs well compared to the baselines in the experiments.

## Weaknesses

- **Technical novelty**: It is a bit unclear as to how novel the proposed methods are. Specifically, both unbiased estimator for RR and the idea of multi-stage training has been investigated pretty extensively in previous work on label-DP (e.g. [Ghazi et al.](https://openreview.net/forum?id=RYcgfqmAOHh) cited in the paper or [Malek et al.](https://openreview.net/forum?id=sR1XB9-F-rv)).
- **Insufficient experimental supports**: Given that the paper is mainly empirical, the experiments do not seems sufficient. For example, in the original submission, there is no comparison against DP-SGD or RLHF. Although these are somewhat addressed in the revision, given that the changes are significant (~25% of the main text), it might be better for the paper to be get another round of proper end-to-end review.
- **Presentation**: The writing can be improved. For example, there are typos in both of the main definitions (Definitions 2 & 3; $y^{2}_i$ vs $y^i_2$) and, given how similar these two definitions are, their differences are not clearly explained. Moreover, the motivation behind protecting the preferences should be expanded, preferably with several clear scenarios.

## Recommendation

Although this paper makes some interesting contribution on an interesting problem, it could benefit from improvement in presentation and additional experiments. Given this, we recommend rejection for now.

**Additional Comments On Reviewer Discussion:**

As briefly mentioned in the meta-review, the original paper contains very thin and clearly insufficient experiments. The main result of the rebuttal is that the authors have added a significant amount of experiments, including with the new baselines and new datasets. Although this seems like they could be efficient and has raised the average score significantly (from 3.75 to 5.25), the changes are quite significant (~25-30% of the main text) and I think they should be properly reviewed in a full reviewing process.

---

### Decision · Program_Chairs · 2025-01-22

Reject